# RepSpec: Structural Re-parameterized Draft Model Training for Speculative Decoding

**Feiye Huo**[1,2,*]  **Jianchao Tan**[2,*]  **Jiahao Liu**[2]  **Zixu Jiang**[2]  **Jiacheng Li**[2]

**Jingang Wang**[2]  **Xunliang Cai**[2]  **Shengli Sun**[1,†]

## Abstract

As the parameter size of large language models (LLMs) continues to grow, the latency of autoregressive inference increases due to memory-bound computational inefficiency. To address this, speculative decoding has been proposed, where a large target model verifies multiple tokens generated in parallel by a smaller draft model. However, the performance of speculative decoding is fundamentally limited by the capacity of the draft model, which is due to the parameter gap between the two models. To overcome this limitation, we propose RepSpec, which combines structural re-parameterization with draft model training. During training, redundant linear structures are introduced and then merged into the backbone network during inference, enhancing the training effectiveness of the draft model without increasing inference cost. By applying our method to improve the current state-of-the-art approach, EAGLE, we achieve a significant improvement in accepted sequence length. Furthermore, considering the specific characteristics of the speculative decoding scenario, we explore a hybrid training strategy that combines linear and non-linear structures, yielding a further improvement in acceptance length.

## 1 Introduction

Large Language Models (LLMs) (Achiam et al., 2023; Touvron et al., 2023; Bai et al., 2023) have demonstrated impressive performance across a wide range of tasks. However, they encounter substantial challenges in autoregressive token generation, primarily due to the high demands on memory bandwidth and suboptimal utilization of GPU resources (Patterson, 2004; Shazeer, 2019), since generating each token necessitates accessing the entire set of model parameters (Radford et al., 2019; Brown et al., 2020). To mitigate these limitations, Speculative Decoding (SD) (Chen et al., 2023; Leviathan et al., 2023) has emerged as a promising solution. SD accelerates inference by generating several candidate tokens using a smaller draft model and validating them in parallel with the larger target model, thereby enhancing GPU computational efficiency. This technique has already been integrated into some of the most recent and influential LLMs (Liu et al., 2024; Team, 2025; Team et al., 2025).

Apart from rule-based approaches that do not require a draft model (Fu et al., 2024; Saxena, 2023; Hu et al., 2024), speculative decoding offers two main paradigms for draft model selection. The first is a training-free approach, where the draft model is constructed by directly extracting a substructure from the target model, commonly referred to as self-speculative decoding (Jun Zhang, 2023). The second paradigm pairs the target model with a separately trained, external smaller model as the draft model, forming a two-model system that requires dedicated training for the draft component (Li et al., 2024a; Cai et al., 2024). Rule-based approaches are generally limited to specific scenarios, while training-free self-speculative decoding typically yields shorter accepted sequence lengths compared to standard dual-model speculative decoding in most cases. Therefore, this paper primarily focuses on dual-model speculative decoding.

---

[*]Equal contribution; [†]Corresponding Author: `slsun@ss.pku.edu.cn`; [1]Peking University; [2]Meituan;

There are two main directions for optimizing dual-model speculative decoding. The first focuses on improving the sampling strategy (Miao et al., 2024; Li et al., 2024b), aiming to maximize the recall of correct candidate tokens generated by the draft model within a limited verification budget. However, such optimizations are still constrained by the inherent capacity of the draft model. Therefore, the second direction is to enhance the capability of the draft model itself, either by modifying its architecture (Li et al., 2024a) or adopting improved training strategies (Zhang et al., 2025; Li et al., 2025). The primary focus of this work is on optimizing the training strategy of the draft model.

For dual-model speculative decoding methods that require training, the limited parameter size of the draft model is a key factor constraining its upper-bound performance. For example, EAGLE uses a single decoder layer from the target model as the starting point for training. While increasing the size of the draft model can indeed improve its performance, it also leads to higher inference costs. Simply adding more layers to the draft model results in a trade-off that is often not worthwhile. Therefore, a natural question arises: Is it possible to temporarily expand the parameter size of the draft model during training to achieve better performance, without introducing any additional inference cost?

To achieve this goal, we introduce the concept of structural re-parameterization into the training of draft models for speculative decoding. The original motivation of structural re-parameterization is to alleviate the vanishing gradient problem in simple deep models like VGG by adding multi-branch topological bypasses similar to those in ResNet (He et al., 2015; Ding et al., 2019; 2021). Due to the additivity of convolution, these additional branches can be merged with the main backbone during inference and thus incur no extra inference cost. Although single-layer decoder draft models typically do not suffer from vanishing gradients, the principle of decoupling the model structure during training and re-coupling it during inference remains highly relevant and beneficial in our setting.

We propose RepSpec, which replaces each linear module in the draft model with multiple redundant linear modules during training, thereby enhancing the training effectiveness. At inference time, these modules can be merged into a single linear layer, ensuring no additional inference cost. Unlike previous works, considering the unique characteristics of speculative decoding—namely, that a modest increase in the draft model's computational cost can be justified if it leads to a substantial improvement in accepted sequence length and overall speedup—we go beyond vanilla structural re-parameterization. Specifically, we explore a hybrid linear–nonlinear training strategy, which further improves the performance of the draft model.

In summary, the contributions of this work are as follows:

- We propose RepSpec, a novel training framework for draft model training in speculative decoding, which leverages structural re-parameterization to enhance training effectiveness without increasing inference cost.
- We go beyond vanilla structural re-parameterization by introducing hybrid linear–nonlinear training strategies, resulting in further performance gains in speculative decoding.
- Experimental results indicate that, after integrating RepSpec, various methods, including the state-of-the-art EAGLE-3, achieve improved training performance in multiple benchmarks and models. On larger target models, employing hybrid structural re-parameterization further yields end-to-end speed gains.

## 2 PRELIMINARIES

### 2.1 SPECULATIVE DECODING

Speculative decoding (SD) (Chen et al., 2023; Leviathan et al., 2023) accelerates model inference by leveraging the parallelism of attention mechanisms. It involves two stages: (1) a draft phase, where a smaller draft model $M_d$ generates draft tokens, and (2) a verification phase, where a larger target model $M_t$ verifies these tokens in parallel.

Let $t_i$ denote the $i$-th token and $T_{s:e} = \{t_s, t_{s+1}, \ldots, t_e\}$ a subsequence, with $T_{1:n}$ as the prefix. For each $t_i$, let $\hat{p}_i$ and $p_i$ be the generation probability from $M_d$ and the verification probability from $M_t$, respectively. Define $\hat{P}_{s:e} = \{\hat{p}_s, \ldots, \hat{p}_e\}$ and $P_{s:e} = \{p_s, \ldots, p_e\}$ as the sequences of these probabilities. Assume the draft model $M_d$ has autoregressively generated $k$ tokens $\hat{T}_{n+1:n+k}$ and

obtained $\hat{P}_{n+1:n+k}$. We then input $\{t_n, \hat{T}_{n+1:n+k}\}$ into $M_t$ to obtain $P_{n:n+k}$ (assuming the prefix's KV cache is available). For each $j = n, \ldots, n + k$, we greedily accept the token if $\arg\max(p_j)$ matches $\hat{t}_{j+1}$, or alternatively, set the acceptance probability to $\min(1, p_j/\hat{p}_j)$. If accepted, the process proceeds to the next token; otherwise, a new token is sampled from a distribution proportional to $\text{norm}(\max(0, p_j - \hat{p}_j))$ and the process repeats. If all candidate tokens are accepted, $p_{n+k}$ is used to sample $t_{n+k+1}$.

## 2.2 STRUCTURAL RE-PARAMETERIZATION

Structural re-parameterization has been widely adopted in convolutional neural networks to decouple model architectures between training and inference. Notably, ACNet (Ding et al., 2019) and RepVGG (Ding et al., 2021) leverage this technique to enhance model capacity during training while maintaining inference efficiency.

In RepVGG, each convolutional block during training consists of multiple parallel branches, such as a $3 \times 3$ convolution, a $1 \times 1$ convolution, and an identity mapping. The overall output is the sum of the outputs from all branches: $y = (K_{3\times3} * x) + (K_{1\times1} * x) + x$, where $K_{3\times3}$ and $K_{1\times1}$ are the kernels for the $3 \times 3$ and $1 \times 1$ convolutions, respectively, $*$ denotes the convolution operation, and $x$ is the input. After training, these branches can be merged into a single $3 \times 3$ convolution by appropriately padding and summing the kernels, exploiting the additivity property of convolution $K_{\text{merged}} = K_{3\times3} + \text{pad}(K_{1\times1}) + \text{pad}(I)$, where $\text{pad}(\cdot)$ denotes zero-padding to match the kernel size, and $I$ is the identity kernel.

Through this approach, the network benefits from richer representations and improved optimization during training, while preserving a simple and efficient structure for inference.

## 3 REPSPEC

To enhance the training of draft models in speculative decoding, we propose **RepSpec**, a novel draft model training approach that incorporates structural re-parameterization.

### 3.1 PURE LINEAR METHOD

The most straightforward manner to apply structural re-parameterization to draft-model training is to augment every eligible linear block with mergeable redundant branches. For a compact Transformer decoder, such blocks include the embedding layer, the projection layers within self-attention, and the linear layers inside the MLP submodule. Empirically, re-parameterizing *all* linear layers in both attention and MLP components is necessary and sufficient.

Denote the original linear layer by Main. During training, we insert (i) a stack of $n$ sequential layers $\{\text{Pre}_i\}_{i=1}^n$ *before* Main, (ii) a stack of $m$ sequential layers $\{\text{Post}_j\}_{j=1}^m$ *after* Main, and (iii) $k$ parallel bypass layers $\{\text{Bypass}_l\}_{l=1}^k$. The augmented mapping is

$$\text{Aug} = \text{Post}_{m:1} \circ \left(\text{Main} + \sum\nolimits_{l=1}^k \text{Bypass}_l\right) \circ \text{Pre}_{1:n}. \tag{1}$$

Let $W_\bullet, b_\bullet$ be the weight and bias of the corresponding module. With the shorthand

$$W_{\text{pre}} = \prod_{i=n}^1 W_{\text{pre}_i}, \quad b_{\text{pre}} = \sum_{i=1}^n \left(\prod_{j=n}^{i+1} W_{\text{pre}_j}\right) b_{\text{pre}_i}, \tag{2}$$

and analogous expressions for $W_{\text{post}}, b_{\text{post}}$, the overall transformation reads

$$\begin{aligned} y = &W_{\text{post}} \left(W_{\text{main}} + \sum\nolimits_l W_{\text{bypass}_l}\right) W_{\text{pre}}\, x \\ &+ W_{\text{post}} \left(W_{\text{main}} + \sum\nolimits_l W_{\text{bypass}_l}\right) b_{\text{pre}} \\ &+ W_{\text{post}} \left(b_{\text{main}} + \sum\nolimits_l b_{\text{bypass}_l}\right) + b_{\text{post}}. \end{aligned}$$

After training, the entire branch-wise structure is *losslessly merged* into a single linear layer New:

$$W_{\text{new}} = W_{\text{post}}\Big(W_{\text{main}} + \sum_l W_{\text{bypass}_l}\Big)W_{\text{pre}},$$

$$b_{\text{new}} = W_{\text{post}}\Big(W_{\text{main}} + \sum_l W_{\text{bypass}_l}\Big)b_{\text{pre}}$$
$$+ W_{\text{post}}\Big(b_{\text{main}} + \sum_l b_{\text{bypass}_l}\Big) + b_{\text{post}}.$$

Because no non-linearity is present, any deeper linear expansion is *functionally equivalent* to the above compact form; excessive depth only increases training cost without improving expressiveness.

By initializing Bypass to zero and Pre/Post to the identity, we ensure that the augmented model is functionally equivalent to the original model at initialization (**Identity Initialization**), while still benefiting from the improved optimization landscape introduced by re-parameterization during training.

The overall process is illustrated in Figure 1.

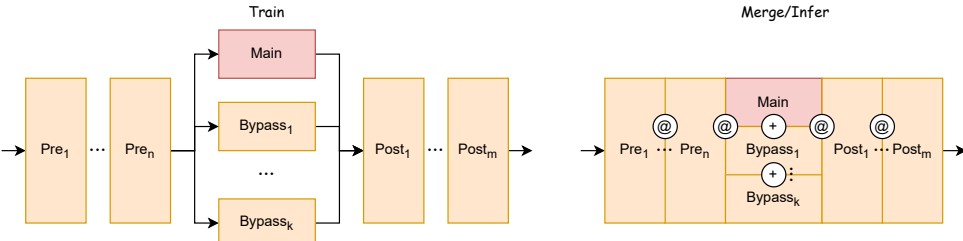

Figure 1: Illustration of the RepSpec method, showing both the training and inference phases. During training (left), inputs pass through a sequence of Pre-layers, then either the Main path or multiple Bypass paths, and finally through Post-layers. In the inference phase (right), the weights of the Main and Bypass layers are first summed element-wise. This summation is then sequentially multiplied by the weights of the Pre and Post layers to yield the final equivalent weights. This approach allows the model to leverage the benefits of multiple training paths without incurring additional computational costs during inference.

## 3.2 HYBRID METHOD

For the pure linear method, the absence of nonlinear activation functions results in a clear limitation on performance improvement. To address this, we consider introducing a minimal number of nonlinear modules to achieve maximal additional gains, even though these nonlinear components cannot be merged during inference. Vanilla structural re-parameterization has never considered nonlinear structures, as standard inference is an isolated process where architectural redundancy directly increases overall computational cost. However, in the context of speculative decoding, the draft model's inference time constitutes only a small fraction of the total decoding time. This allows us to tolerate a slight increase in the draft model's inference cost, provided it leads to a significant improvement in acceptance length and thus yields an overall speedup.

Inserting additional activation functions increases the number of non-mergeable segments and thus the inference cost. Our experiments show that most of the performance gain is achieved by inserting just one activation function; further insertions yield diminishing returns. Therefore, we consider inserting a single nonlinear module, following the pure linear design, at the Pre, Post, or Bypass positions relative to the Main linear layer. Each nonlinear module is implemented as two decomposed linear layers with an activation function in between, using a LoRA-like bottleneck to minimize the number of non-mergeable parameters. Since these modules are still bounded by linear layers, partial merging with other linear components remains possible.

When the activation function is inserted in either the Pre or Post position, the entire linear system is divided into two non-mergeable linear segments. Suppose the Main layer has input dimension in_feature and output dimension out_feature, and the segment interrupted by the activation function

has an intermediate dimension mid_feature. The more complex linear segment can be merged according to the procedure described in Section 3.1, ultimately resulting in two linear layers. When the activation function is inserted in a Bypass branch, it breaks one linear layer into two linear segments separated by the activation function. These two linear segments can be merged with the Pre and Post layers, respectively. However, in this case, the first linear layer can only be concatenated with the Main layer rather than directly summed, resulting in lower inference efficiency compared to inserting the activation function in the Pre or Post positions. Our experiments demonstrate that the optimal hybrid structure is to apply a LoRA-like nonlinearity to the Bypass branch.

# 4 EXPRIMENT

**Models:** We used LLaMA-3.1-Instruct 8B, LLaMA-2-Chat 13B (Touvron et al., 2023) and Vicuna-1.3 7B (Chiang et al., 2023) as target model $M_t$.

**Tasks:** To compare with EAGLE-2 (Li et al., 2024b), we aligned with it on the dataset. For tasks such as multi-round dialogue, code generation, mathematical reasoning, instruction following, summarization, and Q&A, we selected the MT-bench (Zheng et al., 2023), HumanEval (Chen et al., 2021), GSM8K (Cobbe et al., 2021), Alpaca (Taori et al., 2023), CNN/Daily Mail (Nallapati et al., 2016), and Natural Questions (Kwiatkowski et al., 2019), respectively.

**Metrics:** We mainly focus on the following indicators:

- **Decoding speed** $v$**:** The total number of new tokens divided by wall-clock time.

- **Acceptance length** $\tau$**:** The average length accepted by $M_t$ for each generation. Since the $\tau$ includes the token generated by the target model, which is always accepted, the percentage gain should be calculated by subtracting 1 from $\tau$ before computing the improvement.

- **Acceptance rate** $n$-$\alpha$**:** The acceptance rate at the n-th step of autoregressive inference for $M_d$ each generation. Here, we only consider the case of sequential speculative decoding, meaning that this acceptance rate specifically refers to the probability that, at each step, $M_d$'s most probable candidate token is accepted.

**Comparison:** We evaluate the improvements of our method when combined with the widely recognized industry baselines EAGLE-1/3 (Li et al., 2024a;b; 2025), Medusa (Cai et al., 2024), and Hydra (Ankner et al., 2024).

**Environment:** All training experiments were conducted on 8 NVIDIA A100 GPUs (80GB), and all inference experiments were performed on 2 NVIDIA A100 GPUs (80GB).

## 4.1 EFFECTIVENESS

We first evaluate the inference effectiveness of three approaches—standard training (Baseline), pure linear structural re-parameterization (Linear), and hybrid structural re-parameterization (Hybrid)—across models of different sizes and various benchmarks. The experiments primarily focus on the widely used EAGLE-1 and EAGLE-3 methods, with inference performed in both chain mode and the dynamic tree mode of EAGLE-2. All experiments use a temperature of 0. For both re-parameterization methods, we adopt the optimal architectures identified in our ablation studies (see Section 4.2). The results are summarized in Table 1.

On the LLaMA-3.1 8B model, the performance ranks as Linear > Hybrid > Baseline. The pure linear method improves acceptance length without additional inference cost, while the hybrid method achieves greater gains in acceptance length but is slightly slower due to the inclusion of non-mergeable nonlinear components. On the larger 13B model, Hybrid outperforms the others, as the relative draft time decreases and the increased draft cost is offset by larger gains in acceptance length. Specifically, on LLaMA-3.1 8B, Linear accelerates EAGLE-1 by 7%–10% and EAGLE-3 by 4%–6%; on LLaMA-2 13B, Hybrid accelerates EAGLE-1 by 5%–9%.

Beyond the EAGLE family, we also evaluated Medusa and Hydra on Vicuna-1.3 7B. The Linear method delivered the best acceleration, with speedups of 5% for Medusa and 8% for Hydra.

Table 1: Inference performance of the standard training method (Baseline), the pure linear method (Linear), and the hybrid method (Hybrid) across different models and benchmarks, and temperature=0. Here, $\tau$ denotes the acceptance length, $v$ denotes the inference speed (tokens/s), L31 8B refers to LLaMA-3.1-Instruct-8B, L2-13B refers to LLaMA-2-Chat-13B, V 7B refers to Vicuna-1.3 7B, E1/3 refers to EAGLE-1/3, M refers to Medusa, H refers to Hydra, Chain denotes sequential speculative decoding with draft length 5. For EAGLE-1 and EAGLE-3, Tree refers to the dynamic tree method of EAGLE-2. For Medusa and Hydra, Tree indicates the default static tree method. All parameters follow their default configurations.

| Target | Strategy | Draft | Method | MT | | GSM8k | | Alpaca | | Human | | QA | | Sum | | Avg | |
|---|---|---|---|---|---|---|---|---|---|---|---|---|---|---|---|---|---|
| | | | | $\tau$ | $v$ | $\tau$ | $v$ | $\tau$ | $v$ | $\tau$ | $v$ | $\tau$ | $v$ | $\tau$ | $v$ | $\tau$ | $v$ |
| L31 8B | Chain | E1 | Baseline | 2.54 | 48 | 2.74 | 52 | 2.41 | 46 | 2.99 | 58 | 2.28 | 42 | 2.28 | 42 | 2.54 | 48 |
| | | | Linear | 2.67 | **52** | 2.93 | **56** | 2.54 | **48** | 3.22 | **63** | 2.38 | **44** | 2.45 | **46** | 2.70 | **52** |
| | | | Hybrid | **2.79** | 52 | **3.00** | 53 | **2.68** | 47 | **3.39** | 60 | **2.48** | 45 | **2.56** | 45 | **2.82** | 50 |
| | | E3 | Baseline | 3.33 | 65 | 3.32 | 65 | 3.44 | 68 | 3.99 | 74 | 2.96 | 57 | 3.03 | 58 | 3.35 | 64 |
| | | | Linear | 3.58 | **68** | 3.60 | **69** | 3.75 | **74** | 4.25 | **78** | 3.08 | **59** | 3.30 | **61** | 3.60 | **68** |
| | | | Hybrid | **3.62** | 65 | **3.65** | 66 | **3.78** | 70 | **4.32** | 75 | **3.15** | 55 | **3.42** | 59 | **3.66** | 65 |
| | Tree | E1 | Baseline | 3.91 | 70 | 4.17 | 77 | 3.83 | 73 | 4.52 | 82 | 3.46 | 62 | 3.25 | 57 | 3.86 | 70 |
| | | | Linear | 4.16 | **74** | 4.42 | **81** | 4.11 | **75** | 4.82 | **88** | 3.65 | **67** | 3.46 | **62** | 4.10 | **75** |
| | | | Hybrid | **4.34** | 72 | **4.62** | 79 | **4.28** | 71 | **5.05** | 83 | **3.87** | 65 | **3.68** | 60 | **4.31** | 72 |
| | | E3 | Baseline | 5.94 | 100 | 5.86 | 95 | 6.34 | 110 | 6.51 | 109 | 5.13 | 82 | 5.37 | 89 | 5.86 | 98 |
| | | | Linear | 6.00 | **102** | 5.95 | **99** | 6.48 | **115** | 6.60 | **114** | 5.25 | **87** | 5.41 | **91** | 5.95 | **101** |
| | | | Hybrid | **6.08** | 98 | **6.01** | 95 | **6.61** | 114 | **6.65** | 110 | **5.32** | 85 | **5.55** | 90 | **6.03** | 99 |
| L2 13B | Chain | E1 | Baseline | 2.87 | 51 | 3.01 | 56 | 2.75 | 51 | 3.41 | 62 | 2.51 | 48 | 2.65 | 46 | 2.87 | 52 |
| | | | Linear | 2.97 | 54 | 3.09 | 58 | 2.84 | 54 | 3.56 | 64 | 2.60 | 50 | 2.73 | 49 | 2.97 | 55 |
| | | | Hybrid | **3.09** | **56** | **3.20** | **60** | **2.96** | **55** | **3.78** | **67** | **2.73** | **53** | **2.93** | **51** | **3.12** | **57** |
| | Tree | E1 | Baseline | 4.38 | 76 | 4.64 | 80 | 4.32 | 78 | 5.09 | 89 | 3.94 | 70 | 3.48 | 63 | 4.31 | 76 |
| | | | Linear | 4.54 | 79 | 4.75 | 81 | 4.47 | 80 | 5.21 | 92 | 4.10 | 72 | 3.77 | 66 | 4.47 | 78 |
| | | | Hybrid | **4.71** | **82** | **4.95** | **84** | **4.65** | **82** | **5.42** | **93** | **4.25** | **75** | **4.02** | 66 | **4.67** | **80** |
| V 7B | Tree | M | Baseline | 2.38 | 44 | 2.66 | 48 | 2.40 | 45 | 2.75 | 50 | 2.12 | 37 | 2.14 | 38 | 2.41 | 44 |
| | | | Linear | 2.55 | **47** | 2.72 | **51** | 2.45 | **47** | 2.88 | **53** | 2.25 | **40** | 2.26 | **40** | 2.52 | **46** |
| | | | Hybrid | **2.62** | 44 | **2.89** | 49 | **2.51** | 43 | **2.92** | 51 | **2.38** | 37 | **2.34** | 37 | **2.61** | 44 |
| | | H | Baseline | 3.66 | 60 | 3.62 | 53 | 3.50 | 47 | 3.88 | 58 | 2.91 | 40 | 2.85 | 39 | 3.59 | 49 |
| | | | Linear | 3.89 | **64** | 3.92 | **56** | 3.86 | **53** | 4.03 | **60** | 2.96 | **42** | 2.94 | **43** | 3.60 | **53** |
| | | | Hybrid | **4.05** | 63 | **3.99** | 54 | **4.02** | 51 | **4.12** | 59 | **3.02** | 38 | **2.99** | 39 | **3.70** | 51 |

## 4.2 ABLATION STUDY

All experiments in this section are conducted using EAGLE-1 on LLaMA-3.1 8B and MT-bench, with chain mode inference and a draft length of 5.

### 4.2.1 PURE LINEAR STRUCTURE

For the pure linear structure, the initial ablation study evaluates which modules in the Transformer decoder benefit most from structural re-parameterization. We insert a single Bypass into the embedding layer, the q, k, v, o projections in the attention layers, and the gate, up, and down projections in the MLP layers, both individually and in combination, and assess n-$\alpha$ on the validation set. As shown in Figure 2a, the attention and MLP layers are the optimal targets for re-parameterization. Notably, while re-parameterizing the embedding layer provides some evaluation improvement, it can degrade out-of-domain (OOD) performance and, in some benchmarks, even underperform the baseline. Inference results are reported in Table 2.

After identifying the optimal modules for structural re-parameterization (Attn + MLP), we conduct ablation studies on different combinations of linear blocks—Pre, Post, and Bypass—inserted before,

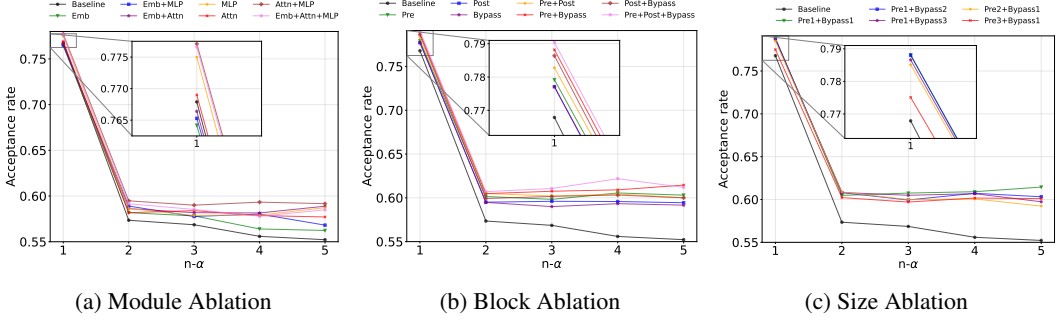

(a) Module Ablation      (b) Block Ablation      (c) Size Ablation

Figure 2: Ablation studies for the pure linear structure method. (a) Structural re-parameterization applied to different modules in the decoder. (b) Ablation of different types and combinations of linear blocks at the optimal insertion position from (a). (c) Ablation of different size configurations for structural re-parameterization based on the optimal position and blocks combination from (a) and (b), the numbers indicate the number of augmented blocks; for example, Pre1+Bypass1 denotes the addition of one Pre and one Bypass block.

Table 2: Comparison of acceptance lengths for ablation study on structural re-parameterization of the embedding layer.

|  | MT Bench | GSM8k | Alpaca | Humaneval | QA | Sum |
|---|---|---|---|---|---|---|
| Baseline | 2.51 | **2.74** | 2.41 | **2.99** | **2.27** | 2.28 |
| Embedding | **2.55** | 2.70 | **2.47** | 2.97 | 2.25 | **2.30** |

after, and in parallel with the original linear layer, respectively. As shown in Figure 2b, while the combination of all three blocks achieves the best training performance, our benchmarking demonstrates that using only Pre and Bypass yields lower training cost and superior empirical results in benchmarks.

After identifying the optimal module positions and block combinations for structural re-parameterization, we further conduct ablation studies on the size of the current configuration. Since, in theory, the pure linear structure can be infinitely expanded, we experiment with increasing the number of Pre and Bypass blocks in the linear layers of both the attention and MLP modules. The experimental results are shown in Figure 2c. It can be observed that using only a single Pre and a single Bypass already constitutes the optimal pure linear structure; adding further redundant structures not only increases the training cost, but may even degrade training performance.

In summary, our experiments show that the optimal pure linear structural re-parameterization method is to insert one Pre and one Bypass into each of the q, k, v, o projections of the attention layers and the gate, up, and down projections of the MLP layers.

### 4.2.2 HYBRID STRUCTURE

In the ablation experiments of the pure linear method, we found that unrestricted increases in linear structural complexity do not result in unlimited performance gains. This naturally motivates us to explore the insertion of nonlinear activation functions to further enhance model performance. We define insertion as splitting a linear block into two linear layers separated by an activation function. For example, given a linear block $X = \mathrm{Linear}(\mathrm{in\_feature}, \mathrm{out\_feature})$, inserting a ReLU yields a new nonlinear block $X' = \mathrm{Sequential}(\mathrm{Linear}(\mathrm{in\_feature}, \mathrm{mid\_feature}), \mathrm{ReLU}(), \mathrm{Linear}(\mathrm{mid\_feature}, \mathrm{out\_feature}))$.

Our first ablation study investigates activation functions inserted into the Bypass block, with $\mathrm{mid\_feature} = \min(\mathrm{in\_feature}, \mathrm{out\_feature})$. While GeLU achieves the best training results (see Figure 3a), benchmark evaluations show that ReLU offers the best speed and simplicity (see Table 3).

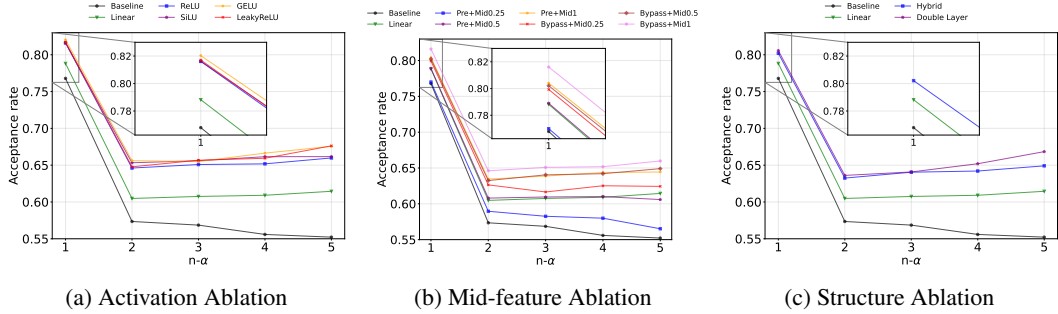

(a) Activation Ablation (b) Mid-feature Ablation (c) Structure Ablation

Figure 3: Ablation studies for the hybrid structure method. (a) Ablation of different activation functions. (b) Using the best activation function, ablation of different insertion blocks, and different intermediate size configurations; for example, Pre+Mid0.5 indicates that the activation function is inserted into the Pre block with mid_feature $= \min(\text{in\_feature}, \text{out\_feature}) \times 0.5$. (c) Ablation of different overall structural designs, where Hybrid denotes the optimal hybrid structural re-parameterization method identified from the previous experiments, and Double Layer denotes simply doubling the number of decoder layers in the EAGLE model.

Table 3: Comparison of acceptance rates and inference speeds for different activation functions

|  | MT Bench | | GSM8k | | Alpaca | | Humaneval | | QA | | Sum | |
| --- | --- | --- | --- | --- | --- | --- | --- | --- | --- | --- | --- | --- |
| Activation | $\tau$ | $v$ | $\tau$ | $v$ | $\tau$ | $v$ | $\tau$ | $v$ | $\tau$ | $v$ | $\tau$ | $v$ |
| ReLU | **2.85** | 51 | 3.05 | 55 | **2.75** | 50 | **3.47** | 62 | **2.54** | 45 | **2.62** | 45 |
| LeakyReLU | 2.84 | 51 | 3.07 | **59** | 2.71 | 48 | 3.43 | 60 | 2.50 | 45 | 2.62 | 45 |
| GeLU | 2.85 | 50 | **3.08** | 56 | 2.75 | 48 | 3.46 | 60 | 2.50 | 43 | 2.62 | 44 |
| SiLU | 2.85 | 50 | 3.06 | 55 | 2.74 | 48 | 3.44 | 58 | 2.51 | 43 | 2.62 | 44 |

The second study examines the effect of inserting ReLU into either the Pre or Bypass block and varying mid_feature. As shown in Figure 3b, using ReLU in the Bypass block with mid_feature $= \min(\text{in\_feature}, \text{out\_feature})$ gives optimal training performance. Reducing mid_feature slightly lowers the acceptance rate but decreases the inference cost. Setting mid_feature $= \min(\text{in\_feature}, \text{out\_feature}) \times 0.5$ achieves the best trade-off, as reported in Table 4.

Table 4: Comparison of acceptance rates and inference speeds for different mid_feature sizes (with a ReLU inserted in the Bypass block).

|  | MT Bench | | GSM8k | | Alpaca | | Humaneval | | QA | | Sum | |
| --- | --- | --- | --- | --- | --- | --- | --- | --- | --- | --- | --- | --- |
| Mid Feature | $\tau$ | $v$ | $\tau$ | $v$ | $\tau$ | $v$ | $\tau$ | $v$ | $\tau$ | $v$ | $\tau$ | $v$ |
| min(in, out) | **2.85** | 51 | **3.05** | 55 | **2.75** | 50 | **3.47** | 62 | **2.54** | 45 | **2.62** | 45 |
| 0.5 * min(in, out) | 2.78 | **53** | 3.00 | **57** | 2.64 | **51** | 3.35 | **63** | 2.46 | **45** | 2.55 | **45** |

The third ablation study compares our approach to other methods that simultaneously increase inference cost and acceptance rate. For example, in EAGLE, the most straightforward approach is to double the number of decoder layers. As shown in Figure 3c, this is compared with the optimal hybrid structural re-parameterization method identified in the previous studies. While doubling the decoder layers yields better training results, our empirical evaluations demonstrate that the hybrid structure achieves superior performance with significantly fewer parameters, as reported in Table 5.

## 4.3 ADDITIONAL OVERHEAD

Although linear structural re-parameterization does not introduce additional overhead during inference, it does incur a non-negligible cost during training. Specifically, RepSpec expands the computational graph by introducing multiple linear branches (Pre, Bypass) as well as optional nonlinear

Table 5: Comparison of acceptance rates and inference speeds for different structures. Hybrid refers to the optimal hybrid structural re-parameterization method, Base refers to only using one decoder layer as the draft model, Double refers to directly using two decoder layers as the draft model. Size refers to the number of trainable parameters.

| Struc. | Size | MT Bench | | GSM8k | | Alpaca | | Humaneval | | QA | | Sum | |
|---|---|---|---|---|---|---|---|---|---|---|---|---|---|
| | | $\tau$ | $v$ | $\tau$ | $v$ | $\tau$ | $v$ | $\tau$ | $v$ | $\tau$ | $v$ | $\tau$ | $v$ |
| Base | 0.24B | 2.51 | 48 | 2.74 | 52 | 2.41 | 46 | 2.99 | 58 | 2.27 | 42 | 2.28 | 42 |
| Double | 0.47B | 2.77 | 49 | 3.00 | 54 | 2.60 | 48 | 3.31 | 60 | 2.42 | 42 | 2.55 | 43 |
| Hybrid | 0.40B | **2.79** | **53** | **3.00** | **57** | **2.68** | **51** | **3.39** | **63** | **2.48** | **45** | **2.56** | **45** |

modules, significantly increasing the computational complexity during training. Here, we provide a systematic analysis of this cost. For Medusa, Hydra, and EAGLE-1, we use native PyTorch distributed training for simple data parallelism (DP). For EAGLE-3, we employ DeepSpeed's ZeRO-3 optimization for training. We specifically record the training speed and GPU memory usage for the Baseline, Linear, and Hybrid methods across different approaches. The experimental results are shown in Table 6.

Table 6: The additional training cost incurred by integrating RepSpec across different methods. All training is conducted in half-precision (FP16). Medusa and Hydra experiments are conducted on Vicuna-1.3 7B, while EAGLE-1 and EAGLE-3 experiments are performed on LLaMA-3.1 8B. Medusa, Hydra, and EAGLE-1 use standard PyTorch distributed training, whereas EAGLE-3 employs DeepSpeed ZeRO-3 optimization. Here, V denotes training speed (steps/s), and MEM denotes GPU memory usage during training (GB).

| | Medusa | | Hydra | | EAGLE-1 | | EAGLE-3 | |
|---|---|---|---|---|---|---|---|---|
| | V | MEM | V | MEM | V | MEM | V | MEM |
| Baseline | 1.42 | 50 | 0.45 | 60 | 1.88 | 28 | 1.29 | 38 |
| Linear | 1.32 | 55 | 0.34 | 76 | 1.57 | 55 | 1.09 | 72 |
| Hybrid | 1.29 | 54 | 0.32 | 73 | 1.54 | 53 | 1.06 | 70 |

In addition to the inevitable training overhead, the Hybrid method also introduces a small amount of forward pass latency during inference. We conducted comparative experiments to assess this aspect, primarily recording GPU memory usage and the latency of each forward pass during inference. The forward pass latency is measured as the average latency over 1000 forward passes under the default inference configuration. The experimental results are shown in Table 7.

Table 7: The additional inference overhead introduced by integrating the Hybrid method across different methods. All experiments are conducted in half-precision (FP16). Medusa and Hydra experiments are conducted on Vicuna-1.3 7B, while EAGLE-1 and EAGLE-3 experiments are performed on LLaMA-3.1 8B. FPL (forward pass latency) denotes the average time per forward pass during inference (ms), and MEM represents GPU memory usage during inference (GB). Hybrid($n$) denotes the Hybrid method with mid_feature $= n \times \min(\text{in\_feature}, \text{out\_feature})$.

| | Medusa | | Hydra | | EAGLE-1 | | EAGLE-3 | |
|---|---|---|---|---|---|---|---|---|
| | FPL | MEM | FPL | MEM | FPL | MEM | FPL | MEM |
| Baseline | 1.38 | 16.4 | 2.04 | 21.5 | 0.87 | 6.0 | 0.95 | 6.4 |
| Hybrid(0.5) | 1.41 | 16.9 | 2.45 | 24.4 | 1.24 | 8.3 | 1.37 | 8.8 |
| Hybrid(1) | 1.44 | 17.2 | 2.61 | 26.8 | 1.57 | 9.0 | 1.64 | 9.8 |

The detailed configurations and re-parameterization designs can be found in Appendix A.

## 5 CONCLUSION

We propose RepSpec, a training-time structural re-parameterization framework designed to enhance draft models for speculative decoding. By inserting mergeable linear branches during training, Rep-Spec increases model capacity without incurring any inference-time overhead; its hybrid variant offers further performance gains with only minimal additional cost. Extensive experiments across various trainable speculative decoding methods, target models, and datasets demonstrate that Rep-Spec consistently improves both acceptance length and end-to-end speed. RepSpec is plug-and-play at training time (requiring only modifications to underlying substructures without altering the overall model design) and can effectively improve the training performance of draft models constrained by parameter count. This flexibility also suggests broad potential for RepSpec's application in other scenarios in the future (including MTP, a pre-training approach based on speculative decoding principles). Our experiments on the 13B model also provide preliminary evidence that the Hybrid method offers greater potential on further larger target models.

## 6 ACKNOWLEDGEMENTS

This work was supported by the Natural Science Foundation of China(62372013), the National Key Research and Development Program of China(2024YFC3307901), and Meituan.

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

## A  IMPLEMENTATION DETAILS

### A.1  TRAINING

All training experiments were conducted on 8 NVIDIA A100 GPUs (80GB).

During the hyperparameter search, we found that when integrating RepSpec ($1 \times$ Bypass + $1 \times$ Pre), doubling the learning rate compared to the baseline typically yields better results. The optimizer and scheduler settings also follow the default configurations. The key training parameters are as follows:

- **Medusa**: learning rate = $2 \times 10^{-3}$, batch size = 8, epochs = 2
- **Hydra**: learning rate = $1 \times 10^{-3}$, batch size = 4, epochs = 10
- **EAGLE-1**: learning rate = $6 \times 10^{-5}$, batch size = 4, epochs = 20
- **EAGLE-3**: learning rate = $1 \times 10^{-4}$, batch size = 2, epochs = 10

Medusa and Hydra also involve the design of the draft model head. We follow the officially released scripts and use the following parameters:

- **Medusa**: `medusa_num_heads` = 3, `medusa_num_layers` = 1
- **Hydra**: `hydra_num_heads` = 4, `hydra_num_layers` = 4

In the Medusa method, there are `medusa_num_heads` Medusa Heads, each consisting of a Sequential block composed of `medusa_num_layers` ResBlocks (square Linear layers with residual connections) and one LM head (a rectangular Linear layer). When integrating RepSpec, we add a Pre and Bypass module to both the Linear layers in the ResBlocks and the LM head.

Hydra is similar to Medusa, with `hydra_num_heads` Hydra Heads, each primarily composed of `hydra_num_layers` ResBlocks and one LM head. The main difference is that each Hydra Head is preceded by a LLaMA2 Decoder, which serves as the embedding layer. When integrating RepSpec, we apply the same structural re-parameterization as in Medusa to the Linear layers in ResBlocks and the LM head. In addition, due to GPU memory constraints, we apply structural re-parameterization only to the Q, K, V, and O projections in the Attention module of the Decoder.

For EAGLE-1/3, as shown in Section 4.2, we perform structural re-parameterization on the Q, K, V, O projections in the Attention module and the gate, up, and down projections in the MLP. Specifically, EAGLE-1 introduces only the Pre and Bypass modules. In contrast, due to the multi-step supervision in EAGLE-3, each training step requires the draft model to perform $n$ (default 7) forward passes. In this case, simple structural re-parameterization can cause output explosion in the later stages of training. To stabilize training, we additionally introduce a Res skip connection module. That is, when in_features = out_feature, direct residual skip connection is adopted; otherwise, a linear block with the shape of (in_feature, out_feature) is added, i.e., $\text{Aug} = \text{Post}_{m:1} \circ \left( \text{Main} + \sum_{l=1}^{k} \text{Bypass}_l \right) \circ \text{Pre}_{1:n} + \text{Res}$. Regarding training data, Medusa, Hydra, and EAGLE-1 are trained on ShareGPT, while EAGLE-3 is trained on both ShareGPT and UltraChat-200k (Ding et al., 2023), with the latter's responses generated by the target model.

The training process of EAGLE-1/3 can be seen in Figure 4. Considering only training performance, Hybrid > Linear > Baseline.

### A.2  INFERENCE

All inference experiments were performed on 2 NVIDIA A100 GPUs (80GB).

During inference experiments, we follow the official implementations of each method. For Medusa and Hydra, we only perform static tree sampling, with the tree structure illustrated in Figure 6 of the Medusa paper.

For EAGLE-1/3, we evaluate RepSpec's performance from two perspectives: In the Chain mode, which is more commonly used in industry, we set the draft length to 5. In the Tree mode, we use the dynamic tree sampling of EAGLE-2, following the official implementation:

- **EAGLE-1**: `depth` = 6, `topK` = 10, `total_token` = 60

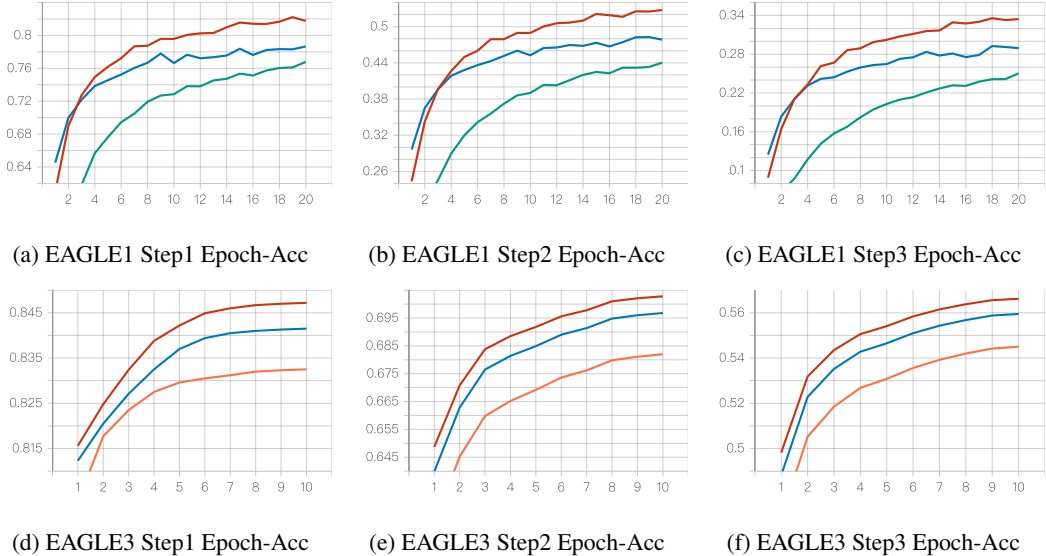

Figure 4: The Epoch-Acc curves for the $n$-th step during training of EAGLE-1/3 with RepSpec on LLaMA-3.1-Instruct 8B, where the vertical axis represents Acc (accuracy) and the horizontal axis represents Epoch. Here, Acc refers to cumulative accuracy, meaning the cumulative Acc at each step is the product of the independent Acc values from all previous steps. Green denotes the EAGLE-1 baseline, orange denotes the EAGLE-3 baseline, blue represents the Linear method, and red represents the Hybrid method combined with RepSpec.

- **EAGLE-3**: `depth = 8`, `topK = 10`, `total_token = 60`

Our evaluation code is built upon SpecBench (Xia et al., 2024).

## B    INITIALIZATION STRATEGY

To guarantee training stability and functional equivalence after merging, we initialize the re-parameterized structures as follows:

- **Main path** (original linear layer): loaded from the pre-trained checkpoint when available; otherwise Xavier uniform.
- **Bypass branches**: $\mathbf{W}_{\text{bypass}} = \mathbf{0}$, $\mathbf{b}_{\text{bypass}} = \mathbf{0}$ so initial forward behaviour equals the main path alone.
- **Pre/Post sequential layers**: identity matrices for weights, zeros for biases, yielding an initial identity transformation.
- **Hybrid nonlinear branch** (LoRA-style): first matrix sampled from $\mathcal{N}(0, 0.02^2)$, second matrix set to zero, giving zero initial output.

This method is referred to as **Identity Initialization**. Consequently, the augmented network starts identically to the original model, avoids early gradient explosion, and produces numerically stable merged parameters after structural re-parameterization.

To validate the effectiveness of this initialization strategy, we set up two commonly used alternative initialization strategies for comparison.

- **Random Initialization**: both the Pre and Bypass modules are initialized with PyTorch's default random initialization.
- **Residual Scaling Initialization**: each re-parameterized branch, including the Pre sequence, is initialized with scaled Gaussian weights whose variance is divided by the total

number of mergeable branches $N$, ensuring that the merged transformation preserves the same activation variance as the original layer.

The experimental results are shown in Figure 5, which indicates that identity initialization > residual scaling initialization > random initialization in terms of performance.

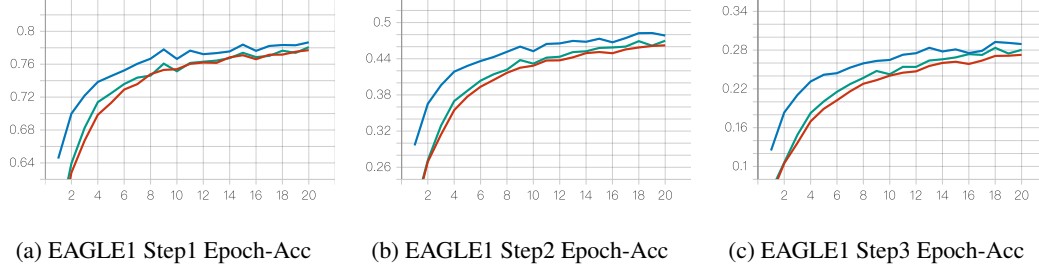

(a) EAGLE1 Step1 Epoch-Acc  (b) EAGLE1 Step2 Epoch-Acc  (c) EAGLE1 Step3 Epoch-Acc

Figure 5: The Epoch-Acc curves for the $n$-th step during training of EAGLE-1 with RepSpec on LLaMA-3.1-Instruct 8B, where the vertical axis represents Acc (accuracy) and the horizontal axis represents Epoch. Here, Acc refers to cumulative accuracy, meaning the cumulative Acc at each step is the product of the independent Acc values from all previous steps. The blue curve represents identity initialization, the green curve represents residual scaling initialization, and the red curve represents random initialization.

## C  WARM START

We also evaluated the effectiveness of RepSpec in the warm-start scenario by using a checkpoint from the EAGLE-1 baseline trained for 20 epochs as the starting point. Since RepSpec introduces new parameters, the optimizer state is reset in our experiments. During warm-start training, RepSpec still follows the identity initialization. The warm-start training process is shown in Figure 6; it can be observed that, although resetting the optimizer leads to fluctuations in the early stages of training, RepSpec ultimately outperforms continued training of the baseline once training stabilizes.

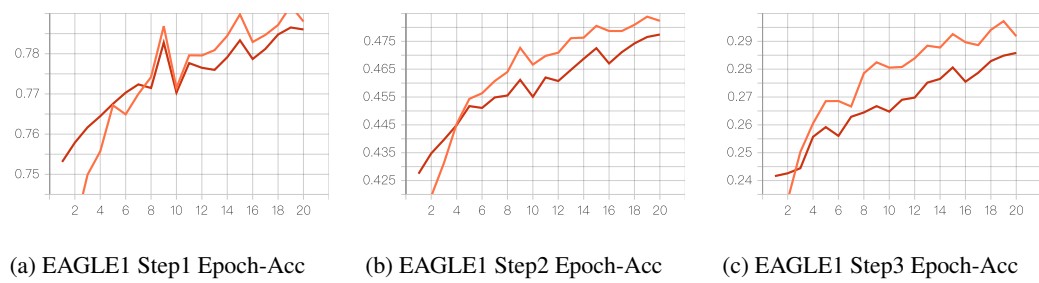

(a) EAGLE1 Step1 Epoch-Acc  (b) EAGLE1 Step2 Epoch-Acc  (c) EAGLE1 Step3 Epoch-Acc

Figure 6: The Epoch-Acc curves for the $n$-th step during training of EAGLE-1 with RepSpec on LLaMA-3.1-Instruct 8B, where the vertical axis represents Acc (accuracy) and the horizontal axis represents Epoch. Here, Acc refers to cumulative accuracy, meaning the cumulative Acc at each step is the product of the independent Acc values from all previous steps. The orange curve represents RepSpec with warm-start initialization, while the red curve represents continued training of the baseline.

## D  SAMPLING RESULTS

Due to space limitations in the main text, we only reported the effectiveness results for temperature $= 0$ in the experimental section. Here, we provide supplementary speculative sampling results for temperature $= 1$. Since Medusa and Hydra are not lossless when temperature $= 1$, we exclude these methods and conduct experiments solely on the EAGLE family. The experimental results are shown in Table 8.

Table 8: Inference performance of the standard training method (Baseline), the pure linear method (Linear), and the hybrid method (Hybrid) across different models and benchmarks, and temperature=1. Here, $\tau$ denotes the acceptance length, $v$ denotes the inference speed (tokens/s), L31 8B refers to LLaMA-3.1-Instruct-8B, L2-13B refers to LLaMA-2-Chat-13B, E1/3 refers to EAGLE-1/3, and Chain denotes sequential speculative decoding with draft length 5. For EAGLE-1 and EAGLE-3, Tree refers to the dynamic tree method of EAGLE-2. All parameters follow their default configurations.

| Target | Strategy | Draft | Method | MT $\tau$ | MT $v$ | GSM8k $\tau$ | GSM8k $v$ | Alpaca $\tau$ | Alpaca $v$ | Human $\tau$ | Human $v$ | QA $\tau$ | QA $v$ | Sum $\tau$ | Sum $v$ | Avg $\tau$ | Avg $v$ |
|---|---|---|---|---|---|---|---|---|---|---|---|---|---|---|---|---|---|
| L31 8B | Chain | E1 | Baseline | 2.52 | 48 | 2.72 | 51 | 2.40 | 46 | 2.98 | 56 | 2.23 | 40 | 2.26 | 41 | 2.52 | 47 |
| | | | Linear | 2.65 | **50** | 2.88 | **54** | 2.51 | **50** | 3.19 | **60** | 2.30 | **43** | 2.43 | **46** | 2.66 | **51** |
| | | | Hybrid | **2.76** | 46 | **2.99** | 53 | **2.68** | 48 | **3.38** | 59 | **2.45** | 42 | **2.57** | 43 | **2.81** | 49 |
| | | E3 | Baseline | 3.32 | 61 | 3.34 | 65 | 3.40 | 61 | 3.73 | 72 | 2.91 | 52 | 3.03 | 58 | 3.29 | 61 |
| | | | Linear | 3.61 | **65** | 3.52 | **70** | 3.77 | **67** | 4.13 | **76** | 3.11 | **59** | 3.27 | **61** | 3.57 | **66** |
| | | | Hybrid | **3.65** | 63 | **3.58** | 66 | **3.82** | 63 | **4.27** | 75 | **3.15** | 55 | **3.33** | 58 | **3.63** | 63 |
| | Tree | E1 | Baseline | 3.50 | 60 | 3.84 | 65 | 3.60 | 64 | 4.27 | 74 | 2.94 | 51 | 3.23 | 56 | 3.56 | 61 |
| | | | Linear | 3.71 | 64 | 4.03 | **70** | 3.73 | 66 | 4.54 | **79** | 3.12 | **53** | 3.42 | **60** | 3.76 | **65** |
| | | | Hybrid | **3.90** | **65** | **4.20** | 67 | **3.92** | 67 | **4.78** | 78 | **3.28** | 51 | **3.67** | 60 | **3.96** | 64 |
| | | E3 | Baseline | 4.44 | 78 | 4.56 | 77 | 4.95 | 80 | 5.27 | 91 | 3.70 | 60 | 4.22 | 70 | 4.52 | 76 |
| | | | Linear | 4.50 | **80** | 4.85 | **83** | 5.03 | 84 | 5.50 | **96** | **3.75** | **63** | 4.45 | **75** | 4.68 | **80** |
| | | | Hybrid | **4.54** | 76 | **4.91** | 80 | **5.18** | **86** | **5.56** | 94 | 3.73 | 56 | **4.46** | 72 | **4.73** | 77 |
| L2 13B | Chain | E1 | Baseline | 2.87 | 51 | 2.98 | 53 | 2.76 | 51 | 3.43 | 63 | 2.53 | 45 | 2.64 | 45 | 2.87 | 51 |
| | | | Linear | 2.94 | 53 | 3.07 | 56 | 2.83 | 53 | 3.53 | 65 | 2.63 | 47 | 2.74 | **50** | 2.96 | 54 |
| | | | Hybrid | **3.07** | 52 | **3.19** | 57 | **3.00** | 55 | **3.77** | 67 | **2.72** | 47 | **2.92** | 49 | **3.11** | 55 |
| | Tree | E1 | Baseline | 4.21 | 71 | 4.65 | 75 | 4.22 | 71 | 4.98 | 85 | 3.94 | 64 | 3.45 | 61 | 4.24 | 71 |
| | | | Linear | 4.39 | 75 | 4.69 | 76 | 4.40 | 76 | 5.09 | 87 | 4.00 | 66 | 3.76 | 65 | 4.39 | 74 |
| | | | Hybrid | **4.57** | **76** | **4.70** | **78** | **4.59** | **78** | **5.28** | **89** | **4.15** | **68** | **4.00** | **67** | **4.55** | **76** |

## E RESULTS ON LARGER MODELS

The following experiments were conducted on SpecForge (Shenggui Li, 2025), with all training and inference parameters following the default configuration. The results of EAGLE-3 combined with RepSpec on LLaMA3.3-70B are shown in Table 9.

Table 9: Inference performance of the standard training method (Baseline), the pure linear method (Linear), and the hybrid method (Hybrid) across different models and benchmarks, and temperature=0. Here, $\tau$ denotes the acceptance length, $v$ denotes the inference speed (tokens/s), L33 70B refers to LLaMA-3.3-Instruct-70B, E3 refers to EAGLE-3. Tree refers to the dynamic tree method of EAGLE-2. All parameters follow their default configurations.

| Target | Strategy | Draft | Method | MT $\tau$ | MT $v$ | GSM8k $\tau$ | GSM8k $v$ | Human $\tau$ | Human $v$ | Avg $\tau$ | Avg $v$ |
|---|---|---|---|---|---|---|---|---|---|---|---|
| L33 70B | Tree | E3 | Baseline | 5.42 | 46 | 6.14 | 55 | 6.32 | 56 | 5.96 | 52 |
| | | | Linear | 5.71 | 49 | 6.40 | 58 | 6.57 | 58 | 6.23 | 55 |
| | | | Hybrid | **5.96** | **50** | **6.56** | **59** | **6.69** | 58 | **6.41** | **56** |

# F WHY PURE-LINEAR RE-PARAMETERIZATION HELPS

## F.1 GRADIENT-VARIANCE REDUCTION: BYPASS

Let the original linear layer be $y = Wx + b$. During training we introduce a *Bypass* branch: $y = (W_0 + W_1)x + (b_0 + b_1)$, where $W_1, b_1$ are initialized to zero. After training we merge $W_{\text{eff}} = W_0 + W_1$, $b_{\text{eff}} = b_0 + b_1$. Both branches share the same upstream gradient $g$; hence

$$g^{(0)} = g, \quad g^{(1)} = g.$$

Adam maintains separate moments for each branch:

$$m_t^{(i)} = \beta_1 m_{t-1}^{(i)} + (1 - \beta_1)g, \quad v_t^{(i)} = \beta_2 v_{t-1}^{((i)} + (1 - \beta_2)g^2.$$

The update for branch $i$ is

$$\Delta W_i = -\alpha \frac{m_t^{(i)}}{\sqrt{v_t^{(i)} + \epsilon}}.$$

Since $v_t^{(1)}$ starts from zero, the Bypass branch receives larger adaptive learning rates early in training, acting as a high-resolution refinement of the main branch.

The total update is

$$\Delta W_{\text{total}} = \Delta W_0 + \Delta W_1 = -2\alpha \frac{1}{2} \left( \frac{m_t^{(0)}}{\sqrt{v_t^{(0)}} + \epsilon} + \frac{m_t^{(1)}}{\sqrt{v_t^{(1)}} + \epsilon} \right).$$

The term in parentheses is the average of two *independently-preconditioned* gradients; its variance is half that of a single branch:

$$\text{Var} \left[ \frac{1}{2} \left( \frac{m_t^{(0)}}{\sqrt{v_t^{(0)}}} + \frac{m_t^{(1)}}{\sqrt{v_t^{(1)}}} \right) \right] = \frac{1}{2}\sigma_{\text{adam}}^2.$$

Thus the effective gradient noise is reduced, leading to stabler updates, larger learning rate and faster convergence.

## F.2 IMPLICIT SPECTRAL REGULARIZATION: PRE/POST

Consider adding *only* Pre and Post factors:

$$y = W_2 W_1 W_0 \, x, \quad \text{with} \quad W_0 \in \mathbb{R}^{p \times q}, \ W_1 \in \mathbb{R}^{r \times p}, \ W_2 \in \mathbb{R}^{q \times r}.$$

Under the balancing effect of gradient flow (Arora et al., 2018), the singular values of the three factors converge to

$$\sigma_j(W_0) \approx \sigma_j(W_1) \approx \sigma_j(W_2) \approx \left[ \sigma_j(W_{\text{eff}}) \right]^{1/3},$$

where $W_{\text{eff}} = W_2 W_1 W_0$ is the *end-to-end* matrix. This balancing behaviour is driven by an implicit minimisation of the log-spectral gaps

$$\sum_j \left| \log \sigma_j(W_1) - \log \sigma_j(W_0) \right|^2 + \sum_j \left| \log \sigma_j(W_2) - \log \sigma_j(W_1) \right|^2,$$

which gradient descent (and Adam) monotonically decrease. As a result, the singular values of $W_{\text{eff}}$ satisfy

$$\sigma_j(W_{\text{eff}}) \approx \sigma_j(W_2) \, \sigma_j(W_1) \, \sigma_j(W_0) \approx \left[ \sigma_j(W_{\text{eff}}) \right]^{1/3} \cdot \left[ \sigma_j(W_{\text{eff}}) \right]^{1/3} \cdot \left[ \sigma_j(W_{\text{eff}}) \right]^{1/3},$$

This implicit regularization penalizes large singular values without any explicit penalty term, and produces a sharper spectral decay for $W_{\text{eff}}$. A faster decay directly reduces the effective rank (Roy & Vetterli, 2007). And smaller effective rank leads to tighter generalization bounds (Bartlett et al., 2017). Hence, our proposed Pre / Post factorization method implicitly shrinks large singular values, accelerates spectral decay, and achieves tighter generalization bounds.

### F.3 SUMMARY

1. Bypass branch averages two independently-preconditioned Adam gradients, reducing variance and speeding up optimization.

2. Pre/Post factorization implicitly balances singular values, encourages low-rank solutions, and enhances generalization.

3. All branches are merged into a single matrix after training, incurring *zero* additional inference cost.

## G  THE USE OF LARGE LANGUAGE MODELS

In the preparation of this paper, we utilized large language models (LLMs) to assist in the following aspects:

- **Language Polishing:** We used LLMs (e.g., GPT-4) to improve the clarity, grammar, and fluency of certain sections of the manuscript. The scientific content, technical contributions, and experimental results were entirely conceived, designed, and verified by the authors.
- **LaTeX Formatting Assistance:** LLMs were used to help generate LaTeX code for tables, figures, and appendix formatting. All content, structure, and data were provided and verified by the authors.

We emphasize that **no novel scientific content or claims were generated by LLMs**. All experiments, analyses, and conclusions were independently conducted and verified by the authors. The use of LLMs was limited to auxiliary tasks to improve presentation efficiency and clarity.

This statement is included in accordance with the transparency guidelines of the conference/journal, and to acknowledge the auxiliary role of LLMs in the writing process.

