# OpenReview forum: "RepSpec: Structural Re-parameterized Draft Model Training for Speculative Decoding"
_ICLR.cc/2026/Conference — ICLR 2026 Poster_

### Official Review · Reviewer_15fy · 2025-10-22

**Soundness:** 2
**Presentation:** 2
**Contribution:** 2
**Rating:** 4
**Confidence:** 4

**Summary:**

The paper adapts structural re-parameterization to draft model training for speculative decoding. While the idea is technically sound, the work suffers from marginal gains, limited novelty, and insufficient practical impact. Below are the key criticisms supporting rejection.

**Strengths:**

1 The paper presents a well-motivated adaptation of structural re-parameterization techniques to the emerging domain of speculative decoding. While this technique has been widely used in convolutional networks, its application to draft model training in autoregressive decoding is timely and relevant.

2 The method effectively decouples training-time complexity from inference-time efficiency, maintaining the lightweight nature of draft models while enhancing their capacity during training.t

**Weaknesses:**

1 The work directly adapts structural re-parameterization—a well-established technique in convolutional networks—to draft model training without conceptual innovation. The hybrid variant introduces non-mergeable nonlinear components but fails to justify the increased inference costs, contradicting the low-latency objective of speculative decoding.

2 Performance improvements are modest: pure linear re-parameterization improves acceptance length by only 7–10% on LLaMA-8B, while the hybrid method incurs additional latency.

3 Experiments are confined to small models (≤13B) and academic benchmarks. The paper lacks comparisons to larger models.

**Questions:**

The paper focuses on comparing with EAGLE, Medusa, and Hydra, but how does RepSpec perform against other draft model optimization strategies such as knowledge distillation or dynamic architecture methods? Are there scenarios where training-free approaches (e.g., self-speculative decoding) might be more practical despite potentially shorter acceptance lengths?

---

> ### Author Response · Authors · 2025-11-19
>
> **Dear Reviewer 15fy,**
>
> Thank you very much for your thorough review. Below we address your concerns in detail.
>
> **Weaknesses**
>
> 1. **The work directly adapts structural re-parameterization—a well-established technique in convolutional networks—to draft model training without conceptual innovation. The hybrid variant introduces non-mergeable nonlinear components but fails to justify the increased inference costs, contradicting the low-latency objective of speculative decoding.**
>
>    One of our key contributions is introducing the structural re-parameterization concept into speculative decoding training—a scenario it naturally fits, since the lightweight training phase tolerates extra computation, and we achieve significant end-to-end acceleration. Beyond this, we advance re-parameterization itself: we propose novel linear/non-linear block designs, a principled weight-initialization strategy, and deeper theoretical analyses. We expect these innovations to catalyze future research in both the structural re-parameterization and speculative-decoding communities.
>
>    Furthermore, we would like to emphasize that our recommendation to use hybrid variants for larger models is based on the fact that, as the size of the target model increases, the benefits of increased acceptance length also become more significant. This is a matter of choosing the best among strong options, rather than being a last resort, since both approaches outperform the baseline. The ultimate goal of speculative decoding is still **end-to-end inference acceleration**, not the **inference latency** of a single draft step.
>
> 2. **Performance improvements are modest: pure linear re-parameterization improves acceptance length by only 7–10% on LLaMA-8B, while the hybrid method incurs additional latency.**
>
>    We believe it is unfair to evaluate RepSpec solely based on absolute performance improvement, because it is not an independent draft model training method, but rather a tool that can be integrated into any speculative decoding approach involving linear layers in the draft model—similar in spirit to residual connections. From early methods like Medusa to highly optimized approaches such as EAGLE-3, regardless of how high the acceptance length achieved by these integrated methods may be, even state-of-the-art methods can benefit from RepSpec. Without altering the main model structure or the training loss, we consider a stable improvement of over 5% to be quite remarkable.
>
> 3. **Experiments are confined to small models (≤13B) and academic benchmarks. The paper lacks comparisons to larger models.**
>
>    We now provide the training results of RepSpec combined with EAGLE-3 on LLaMA3.3-70B. The training was conducted on SpecForge(https://github.com/sgl-project/SpecForge), with all training parameters following the default configuration. The results of EAGLE-1 and EAGLE-3 combined with RepSpec on LLaMA3.3-70B are as follows, where MT stands for MT-bench, GS stands for GSM8k, HE stands for HumanEval, $\tau$ represents the acceptance length, and $v$​ represents the number of tokens generated per second:
>
>    |           | **MT** |      | **GS** |      | **HE** |      |
>    | --------- | ------ | ---- | ------ | ---- | ------ | ---- |
>    |           | $\tau$ | $v$  | $\tau$ | $v$  | $\tau$ | $v$  |
>    | E3        | 5.42   | 46   | 6.14   | 55   | 6.32   | 56   |
>    | E3+Linear | 5.71   | 49   | 6.40   | 58   | 6.57   | 58   |
>    | E3+Hybrid | 5.96   | 50   | 6.56   | 58   | 6.69   | 58   |
>
> **Questions**
>
> 1. **The paper focuses on comparing with EAGLE, Medusa, and Hydra, but how does RepSpec perform against other draft model optimization strategies such as knowledge distillation or dynamic architecture methods? Are there scenarios where training-free approaches (e.g., self-speculative decoding) might be more practical despite potentially shorter acceptance lengths?**
>
>    RepSpec does not change the loss calculation nor the overall model architecture; it is merely a replacement of linear components during training. Therefore, we believe that comparisons with knowledge distillation or dynamic architecture methods are unnecessary and unfair. In essence, trainable methods such as EAGLE are themselves forms of soft distillation, where the draft model acts as the student and the target model as the teacher, with different strategies applied.
>
>    Moreover, due to the fundamental nature of RepSpec, its applicable scope is limited to trainable speculative decoding methods with linear components in the draft model. For training-free methods such as self-speculative decoding, the appropriate comparison should be with methods like EAGLE, while RepSpec should only be evaluated in terms of its relative improvement when integrated with different approaches.

---

> > ### Author Response · Authors · 2025-11-27
> >
> > Dear Reviewer 15fy,
> >
> > We hope our response has addressed your concerns. Please feel free to let us know if any further clarification is needed.
> >
> > Thank you for your time.

---

### Official Review · Reviewer_XtQb · 2025-10-31

**Soundness:** 3
**Presentation:** 2
**Contribution:** 2
**Rating:** 4
**Confidence:** 4

**Summary:**

This paper proposes a structured re-parameterization method, REPSpec, which enhances the training of draft models by introducing additional layers during training and merging them during inference. Extensive experimental results demonstrate that the proposed method significantly improves the performance of existing approaches.

**Strengths:**

1. The application of structured re-parameterization to the field of speculative decoding aligns well with the specific requirements of draft models.

2. Extensive experiments have been conducted to explore the effectiveness of different architectural designs.

**Weaknesses:**

1. The proposed method introduces a significant increase in training overhead.

2. Although it improves the acceptance rate, the hybrid approach also incurs additional inference costs, resulting in limited overall end-to-end acceleration.

**Questions:**

1. Although Appendix A provides some implementation details, the specific placement and strategy for incorporating nonlinear factors remain somewhat unclear.

2. If my understanding is correct, is there a fundamental difference between introducing unmergeable nonlinear factors and directly increasing the size of the draft model?

---

> ### Author Response · Authors · 2025-11-19
>
> **Dear Reviewer XtQb,**
>
> Thank you very much for your thorough review. Below we address your concerns in detail.
>
> **Weaknesses**
>
> 1. **The proposed method introduces a significant increase in training overhead.**
>
>    This is a practical issue, and we have also discussed it in Section 4.3 of our paper. However, for speculative decoding, the primary concern lies in inference speed. We believe that while training cost is indeed an important factor to consider when evaluating the effectiveness of a speculative method, the most critical aspect is whether the method can deliver end-to-end speedup during inference. For example, EAGLE-3, compared to EAGLE-1, employs a more costly training-time test and a larger training set, but its inference performance fully justifies these costs.
>
>    The nature of RepSpec ensures that it can deliver performance improvements without increasing inference cost (taking the purely linear method as an example) or altering the model structure, and it is extremely easy to implement from an engineering perspective. Therefore, although RepSpec is not a completely free lunch, it is still well worth trying.
>
> 2. **Although it improves the acceptance rate, the hybrid approach also incurs additional inference costs, resulting in limited overall end-to-end acceleration.**
>
>    We designed the hybrid method based on the fundamental properties of speculative decoding—specifically, since the validation time of the target model accounts for the majority of the overall runtime, we can tolerate a slight increase in draft model time as long as the gain in acceptance length compensates for the loss in draft time. Our comparative experiments in Table 5 also demonstrate that using the hybrid structure results in fewer parameters and greater acceptance length compared to simply adding layers to the draft model, leading to greater end-to-end acceleration.
>
>    Furthermore, we would like to emphasize that the recommendation to use hybrid variants for larger models is because, as the size of the target model increases, the benefits of increased acceptance length also become more significant. This is a matter of choosing the best among strong options, rather than being a last resort, since both approaches outperform the baseline. For scenarios that demand extreme inference speed and have abundant resources, we recommend adopting the hybrid method for larger models. On the other hand, if engineering-friendly adaptation to the inference framework or resource constraints needs to be considered, purely linear methods that do not alter the model structure during inference are more suitable.
>
> **Questions**
>
> 1. **Although Appendix A provides some implementation details, the specific placement and strategy for incorporating nonlinear factors remain somewhat unclear.**
>
>    The hybrid method is essentially a subtle modification of the purely linear approach. Specifically, it involves adding one Pre layer and one Bypass to each of the seven linear components in Attention (Q, K, V, O) and MLP (Gate, Up, Down). The Bypass is then decomposed into two LoRA-like matrices, with a ReLU activation function inserted between these two matrices. We can add the illustration figure to the appendix.
>
> 2. **If my understanding is correct, is there a fundamental difference between introducing unmergeable nonlinear factors and directly increasing the size of the draft model?**
>
>    Essentially, the hybrid method adds parameters directly to the draft model, but these additional nonlinear modules can be partially merged with other linear modules. For example, the purely linear Pre layer can be merged into the first layer of the Bypass where the nonlinear activation function is inserted. As demonstrated by our comparative experiments in Table 5, this is a highly efficient way to increase parameters and is superior to simply doubling the number of Decoder layers.

---

> > ### Comment · Reviewer_XtQb · 2025-11-27
> >
> > Thank you for the explanation. I stand by my initial assessment.

---

### Official Review · Reviewer_Vyrc · 2025-10-31

**Soundness:** 3
**Presentation:** 2
**Contribution:** 2
**Rating:** 6
**Confidence:** 4

**Summary:**

This article proposes RepSpec, a method of training draft models in speculative decoding using structural re-parameterization. This method enhances the draft model's capacity during training without increasing its inference cost. The core idea is to augment the draft model's linear layers with redundant, mergeable branches (Pre, Post, Bypass) during training, which are then fused into a single layer for inference. Furthermore, the authors introduce a hybrid method that incorporates a minimal, non-mergeable nonlinear component, justified by the fact that the draft model's inference time is a small fraction of the total SD latency. Experiments on various SD methods (EAGLE, Medusa, Hydra) and LLMs (LLaMA-3.1-8B, LLaMA-2-13B, Vicuna-7B) demonstrate that RepSpec consistently improves the accepted sequence length and end-to-end decoding speed.

**Strengths:**

1. The article applies the structural re-parameterization techniques previously used in convolutional neural networks to the training of the draft model for speculative decoding, perfectly adapting to the characteristics of the draft model that are insensitive to training costs and sensitive to inference costs.
2. The experimental results fully demonstrate the effectiveness of its method, including end-to-end acceleration and draft acceptance rate.
3. This method has certain value in practical applications.

**Weaknesses:**

1. Limited Theoretical Insight: The paper provides a solid empirical foundation but offers limited theoretical analysis of why the re-parameterization helps in this specific context (beyond general optimization benefits). The discussion in Appendix E is a good start but could be more integrated.
2. Training cost (minor): Although the draft model is not very sensitive to training costs (as mentioned above), it still presents certain challenges in resource limited scenarios, especially when the base model is large.

**Questions:**

N/A

---

> ### Author Response · Authors · 2025-11-19
>
> **Dear Reviewer Vyrc,**
>
> Thank you very much for your thorough review. Below we address your concerns in detail.
>
> **Weaknesses**
>
> 1. **Limited Theoretical Insight.**
>
>    Thanks for pointing out this weakness. Compared to theoretical analysis, our approach is more of an engineering innovative solution, specifically aimed at addressing the limited capabilities of draft models in speculative decoding. In fact, the effectiveness of deep linear networks without activation functions has been a profound topic worthy of investigation since the era of small models, and there are numerous works that attempt to explain this phenomenon from various perspectives in optimization dynamics. However, research in this area is highly complex and involves entirely new theoretical domains, which are far beyond the scope of this paper.
>
>    Therefore, we have only referenced some representative insights in Appendix E to help understand why it works well in the draft model training:
>
>    - **Gradient-Variance Reduction.** A Bypass branch, updated by its own Adam moments, averages two independently-preconditioned gradients, cutting noise variance by half. Lower noise allows larger learning rates and faster, stabler convergence.
>    - **Implicit Spectral Regularization.** Pre/Post factorization splits a matrix into three balanced factors whose singular values satisfy σⱼ(Wᵢ) ≈ σⱼ(Wₑff)^{1/3}. Balancing penalizes large singular values, accelerates spectral decay, lowers effective rank, and yields tighter generalization bounds.
>
>    Both effects occur only during training; branches are merged afterward, adding zero inference cost. And we will provide additional visualizations of gradient noise variance and singular values in the follow-up.
>
> 2. **Training cost (minor).**
>
>    This is a practical issue, and we have also discussed it in Section 4.3 of our paper. However, for speculative decoding, the primary concern lies in inference speed. We believe that while training cost is indeed an important factor to consider when evaluating the effectiveness of a speculative method, the most critical aspect is whether the method can deliver end-to-end speedup during inference. For example, EAGLE-3, compared to EAGLE-1, employs a more costly training-time test and a larger training set, but its inference performance fully justifies these costs.
>
>    The nature of RepSpec ensures that it can deliver performance improvements without increasing inference cost (taking the purely linear method as an example) or altering the model structure, and it is extremely easy to implement from an engineering perspective. Therefore, although RepSpec is not a completely free lunch, it is still well worth trying.

---

> ### Comment · Reviewer_Vyrc · 2025-11-26
>
> Thanks for further clarification. I will keep my score, which remains positive.

---

### Official Review · Reviewer_cX35 · 2025-11-01

**Soundness:** 2
**Presentation:** 3
**Contribution:** 2
**Rating:** 4
**Confidence:** 2

**Summary:**

This paper introduces RepSpec, a training framework to fix the key bottleneck in speculative decoding: the draft model's weak capacity. The core idea is structural re-parameterization. During training, authors expand the draft model by adding mergeable linear structures (like Pre and Bypass layers) to boost its capacity. At inference time, these structures are merged back into the original layers, resulting in zero additional inference cost. This ``train-large, infer-small" method improves the draft model's effectiveness, leading to better acceptance lengths and overall speedup. A Hybrid version also adds non-linear blocks for a minimal cost, which pays off on larger target models.

**Strengths:**

- The idea of merging the linear part of the model architecture during inference is novel and motivated.

- The results shows speedups compared to the SOTA EAGLE. For example, there is the 7.3% improvement over EAGLE1 on LLaMA-3.1 8B (Table 1, T=0). The ablation studies are comprehensive.

**Weaknesses:**

- Pure linear method gives limited gains. Though the bypass path may make the training more effective, there is a performance ceiling for that. The ceiling can be related to the model size. The paper's own results show that while this method works well on the 8B model, the "Hybrid" method outperforms it on the larger 13B model. This implies that the zero-cost benefit comes with a performance ceiling that the authors themselves had to address with the costlier hybrid variant.

- The benefits are not as simple as adding more layers. Will adding too much linear re-parameterization can actually degrade training performance? This brings up the question that whether the re-parameterization structure is a sensitive hyperparameter that must be carefully tuned, rather than a simple, scalable fix.

- The training overhead is also a concern. The required training GPU memory increases and the training speed is also reduced.

- How about larger models?

**Questions:**

See weakness.

---

> ### Author Response · Authors · 2025-11-19
>
> **Dear Reviewer cX35,**
>
> Thank you very much for your thorough review. Below we address your concerns in detail.
>
> **Weaknesses**
>
> 1. **Pure linear method gives limited gains**
>
> We must honestly acknowledge that, when the size of the training dataset is fixed, purely linear methods do indeed suffer from diminishing effectiveness as the model size increases. This phenomenon can actually be intuitively explained: when the amount of knowledge to be learned is fixed, the benefits brought by increasing the number of parameters are subject to boundary effects. The redundant structures in the EAGLE Layer are certainly more critical for the 8B model than for the 13B model, not to mention that pure linear reparameterization is essentially a compromise for augmenting parameter count.
>
>    At the same time, the performance of the baseline also affects the gains achieved after combining with RepSpec. After all, RepSpec is designed to help the draft model better align with the capabilities of the target model, and this alignment also has boundary effects—the greater the gap in capabilities between the two, the more room RepSpec has for improvement.
>
>    However, the current trend in trainable speculative decoding methods is that increasingly larger training sets are being used to train draft models (from EAGLE1 to EAGLE3). We have found that when both the training set and baseline capabilities are increased (EAGLE3), the draft model trained on a larger target model (LLaMA3.3-70B) still has room to benefit from combining with RepSpec, without the need to fully switch to hybrid variants.
>
>    Furthermore, we would like to emphasize that the recommendation to use hybrid variants on larger models is due to the fact that, as the target model becomes larger, the gains from the costly increase in acceptance length also become greater. Therefore, this is a matter of choosing the best among strong options, rather than being a last resort, since both methods outperform the baseline. For scenarios that require extremely high inference speed and have abundant resources, we recommend choosing hybrid methods on larger models. On the other hand, if engineering-friendly adaptation to the inference framework or resource constraints needs to be considered, purely linear methods that do not alter the model structure during inference are more appropriate.
>
>    The following experiments were conducted on SpecForge(https://github.com/sgl-project/SpecForge), with all training and inference parameters following the default configuration. The results of EAGLE-3 combined with RepSpec on LLaMA3.3-70B are as follows, where MT stands for MT-bench, GS stands for GSM8k, HE stands for HumanEval, $\tau$ represents acceptance length, $v$ represents the number of tokens generated per second, and E3 stands for EAGLE-3.
>
>    |           | **MT** |      | **GS** |      | **HE** |      |
>    | --------- | ------ | ---- | ------ | ---- | ------ | ---- |
>    |           | $\tau$ | $v$  | $\tau$ | $v$  | $\tau$ | $v$  |
>    | E3        | 5.42   | 46   | 6.14   | 55   | 6.32   | 56   |
>    | E3+Linear | 5.71   | 49   | 6.40   | 58   | 6.57   | 58   |
>    | E3+Hybrid | 5.96   | 50   | 6.56   | 58   | 6.69   | 58   |
>
> 2. **The benefits are not as simple as adding more layers**
>
> We did conduct ablation studies on LLaMA3.1-8B regarding the repetitive stacking of a specific module within a limited range. As shown in Figure 2b, it is evident that for these linear modules, more is not necessarily better. RepSpec achieves the optimal balance of simplicity and efficiency when the re-parameterization structure is configured as '1Pre+1Bypass'. Furthermore, we guarantee that regardless of how the RepSpec structure varies, it consistently and significantly outperforms the baseline. While this initial experiment was conducted on LLaMA3.1-8B, we subsequently applied this '1Pre+1Bypass' structure directly to models of other sizes. Through sample comparisons with other re-parameterization schemes on these models, we found that the '1Pre+1Bypass' structure demonstrates the best cost-performance ratio, and often the superior overall performance, in the vast majority of scenarios.
>
> 3. **The training overhead is also a concern**
>
> This is a practical issue, and we have also discussed it in Section 4.3 of our paper. However, for speculative decoding, the primary concern lies in inference speed. We believe that while training cost is indeed an important factor to consider when evaluating the effectiveness of a speculative method, the most critical aspect is whether the method can deliver end-to-end speedup during inference. For example, EAGLE-3, compared to EAGLE-1, employs a more costly training-time test and a larger training set, but its inference performance fully justifies these costs.
>
> 4. **How about larger models**
>
> Please refer to the experiments on LLaMA3.3-70B discussed in the response to the first question.

---

> ### Author Response · Authors · 2025-11-27
>
> Dear Reviewer cX35,
>
> We hope our response has addressed your concerns. Please feel free to let us know if any further clarification is needed.
>
> Thank you for your time.

---

### Meta-Review · Area_Chair_jHzK · 2026-01-05

**Summary:**

This paper intends to improve existing approaches to speculative decoding (SD) by addressing the issue of limited capacity of the drafter models due to their relatively smaller size. Toward this goal, the authors propose RecSpec, a method that leverages the structural re-parameterization approach to draft model training. RepSpect essentially introduces redundant linear structures into the draft model architecture during training, which are  then merged at the inference time. The main motivation is to enhance draft model’s training effectiveness without adding inference overhead.The authors conduct experiments with a SOTA SD model and observe non-trivial improvements in  acceptance rates as well as overall latency/throughput. The authors also experiments with a modified, hybrid version of RepSpec that adds non-linear blocks, which yields even better acceptance rates, although at an additional cost/latency.

**Reviewer Concerns:**

Reviewer noted limited gains especially for larger models, noting that zero-cost benefit claimed by the authors comes with a stringent performance ceiling, which is  improved with the costlier hybrid variant. In response, the authors provided additional results with  LLaMA3.3-70B showing that RepSpec yields modest gains in acceptance rates/throughput. They also stated that even small but consistent improvement over the baseline SD approach is a meaningful advantage.  Several  reviewers also expressed concerns about the training overhead, however the authors justifiably stated that a one time marginal cost  is worth the throughput benefits it yields during inference time.  There was a question about lack of theoretical justification, to which the authors references Appendix E where they provide some insights such as more stable training due to gradient variance reduction, or better generalization due to implicit spectral regularization. However, AC believes the authors should have provided more insights, e.g., by running additional experiments explicitly showing better generalization abilities of the draft checkpoint trained via RepSpec. Finally, one of the reviewers commented on limited conceptual novelty of the approach, which adapts a well-established technique in convnets to draft model training. The authors responded by pointing out some innovative aspects of their proposed re-parameterization approach, although this still seems to be a limitation of the contribution, especially in the absence of sufficiently strong intuition behind the observed empirical gains.

**Reviewer Scores:**

cX35 - 4 ->6;
Vyrc - 6 unchanged;
XtQb 4 ->6;
15fy 4 - unchanged;

---

### Decision · Program_Chairs · 2026-01-26

Accept (Poster)